# NOVEL CLASS DISCOVERY UNDER UNRELIABLE SAMPLING

## ABSTRACT

When sampling data of specific classes (i.e., *known* classes) for a scientific task, collectors may encounter unknown classes (i.e., *novel* classes). Since these novel classes might be valuable for future research, collectors will also sample them and assign them to several clusters with the help of known-class data. This assigning process is also known as *novel class discovery* (NCD). However, sampling errors are common in practice and may make the NCD process unreliable. To tackle this problem, this paper introduces a new and more realistic setting, where collectors may misidentify known classes and even confuse known classes with novel classes - we name it *NCD under unreliable sampling* (NUSA). We find that NUSA will empirically degrade existing NCD methods if taking no care of sampling errors. To handle NUSA, we propose an effective solution, named *hidden-prototype-based discovery network* (HPDN). HPDN first trains a deep network to fully fit the wrongly sampled data, then applies the relatively clean hidden representations yielded by this network into a novel mini-batch K-means algorithm, which further prevents them overfitting to residual errors by detaching noisy supervision timely. Experiments demonstrate that, under NUSA, HPDN significantly outperforms competitive baselines (e.g., $6\%$ more than the best baseline on CIFAR-10) and keeps robust even encountering serious sampling errors.

## 1 INTRODUCTION

Data, algorithms, and computing power create the boom in the field of artificial intelligence, especially the supervised learning with many powerful deep models (Deng et al., 2009; Krizhevsky et al., 2012; Simonyan & Zisserman, 2015). Although these deep models can accurately identify or cluster the classes appeared in the training set (i.e., known/seen classes), they do not have reliable extrapolating ability in front of novel classes (i.e., unseen classes). For young children, after seeing some common vehicles (e.g., cars and bicycles), they can easily distinguish (cluster) the unseen but similar ones (e.g., trains and steamships) based on previous experience. This fact motivates researchers to formulate a novel problem called *novel class discovery* (NCD) (Han et al., 2020; 2019; Hsu et al., 2018; 2019; Zhao & Han, 2021; Zhong et al., 2021a;b), aiming to accurately cluster novel classes using labeled known-class data and unlabeled novel-class data.

Existing work (Chi et al., 2022) demystifies the underlying assumptions of NCD, then define NCD strictly from the perspective of sampling, making NCD problem theoretically solvable. Specifically, given a sampling task (i.e., collecting known-class data), the known-class and novel-class data are sampled in the same scenario, but the novel-class data are sampled passingly, and experts cannot identify them. Since the same scenario indicates that two groups have similar high-level semantic features, employing knowledge of known classes to assist the clustering of novel classes is meaningful.

However, for professional and difficult sampling tasks, the experts may wrongly identify known classes (i.e., internal errors), and even confuse the known classes with novel classes (i.e., external errors). A direct example is to sample different varieties of privet, a type of shrubs. If experts are not very proficient, they may wrongly identify *ligustrum vicaryi* and *ligustrum quihoui* (internal errors), since they look very similar. Furthermore, they may confuse *ligustrum vicaryi* and *kerria japonica* (i.e., external errors), since they both appear to be red. Motivated by this scenario, we propose a new and challenging problem called *NCD under unreliable sampling* (NUSA), where we try to discover novel classes under both internal and external sampling errors, as shown in Figure 1.

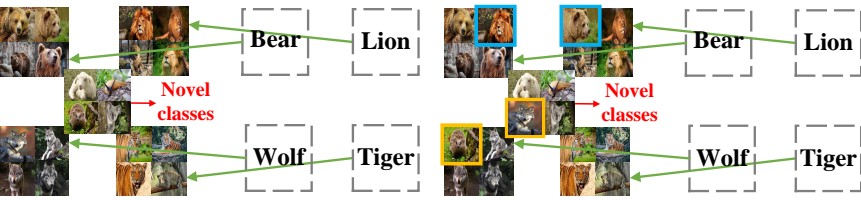

(a) Novel class discovery (NCD).  (b) NCD under unreliable sampling (NUSA).

Figure 1: *Novel class discovery* (NCD, (a)) is formulated by a sampling process (green arrows). When collectors sample the data of required classes (i.e., bear, lion, wolf, and tiger) in a scenario, they may encounter novel classes (i.e., squirrel and hare) that are unfamiliar, and they had better also sample them for future research. Then, assigning them to several clusters with the help of known-class data is known as NCD. However, collectors possibly make mistakes in practice, which is named as *NCD under unreliable sampling* (NUSA, (b)). Here we consider two cases, where they misidentify the known classes (i.e., internal errors, shown in *blue* boxes) and even confuse known classes with novel classes (i.e., external errors, shown in *yellow* boxes).

The most direct solution to NUSA is the existing NCD methods (Fini et al., 2021; Han et al., 2020; 2019; Zhong et al., 2021a), and the results are shown in the left one of Figure 2. Clearly, NUSA empirically degrades the four representative NCD methods, and previous methods cannot handle NUSA well. Moreover, the label-noise learning methods (Han et al., 2018; Li et al., 2020) can be employed to correct the labels[1] of all the sampled data first, and then these data and revised labels will be applied into the existing NCD methods to solve NUSA, which can be regarded as a two-step solution to NUSA. However, existing label-noise learning methods cannot fully eliminate noises, and experimental results (Table 1) show that residual errors still weaken NCD methods. Based on these empirical results, the two types of sampling errors substantially invalidate both NCD methods and the above two-step methods.

To address the sampling errors in NUSA, we propose the *hidden-prototype-based discovery network* (HPDN). In terms of supervision, the sampled data with errors can be treated as data with label noises. Li et al. (2021a) pointed out that if an architecture "suits" one task, training with noisy supervisions can induce useful hidden representations. Inspired by this conclusion, HPDN first trains a deep network (initialized by SimCLR (Chen et al., 2020)) to fully fit the wrongly sampled data. This network can yield relatively clean hidden representations for novel-class data (Li et al., 2021a). However, the right one of Figure 2 indicates the residual errors in hidden representations still degrade the existing NCD methods. This is caused by the strong memory of deep networks (Zhang et al., 2021), leading to the accumulation of residual noisy supervision information in training procedure.

To avoid further errors accumulation in the representation, unlike existing NCD methods, at clustering stage, we detach the noisy supervision information in time. Then, we employ K-means (MacQueen et al., 1967), an unsupervised clustering algorithm. Naive K-means uses all data representations at a time and is sensitive to initial centers. However, given many data representations, proper initialization is hard to choose, and it may be negatively affected by residual errors existed in representations, and furthermore many iterations are required. Thus, we propose the mini-batch K-means with memories of clustering centers (i.e., prototypes) to discover novel classes using hidden representations. Mini-batches are easier to be initialized with K-means++ (Arthur & Vassilvitskii, 2006) due to their smaller sample complexity. After obtaining centers of each batch, we take their matched average value (i.e., prototypes) to initialize each batch in next epoch, taking care of the whole dataset. In this way, prototypes will gradually converge to a stable state as the final clustering centers.

To verify the effectivenss of HPDN, we perform experiments on three benchmarks: CIFAR-10, CIFAR-100 and ImageNet. Experimental results show that HPDN outperforms existing baselines (e.g., 6% more than the best baseline on CIFAR-10) and is very robust to sampling errors in NCD (Figure 4), which confirms the effectiveness of HPDN.

---

[1]The novel classes are currently considered as one class.

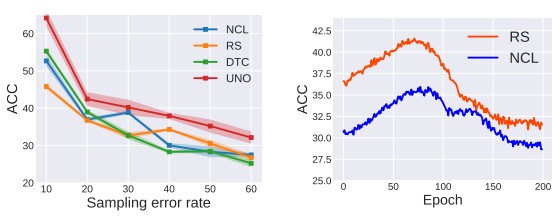

**Left figure**: Four SOTA NCD methods will fail as the sampling error rate increases (from $10\%$ to $60\%$). **Right figure**: The residual sampling errors in hidden representations will accumulate in the training procedure of existing NCD methods (taking RS (Han et al., 2020) and NCL (Zhong et al., 2021a) on CIFAR-10 under $40\%$ error rate as an example), due to the strong memory of deep networks (Zhang et al., 2021). This leads to the ACC dropping quickly at a later stage.

Figure 2: Failures of SOTA NCD methods when encountering unreliable sampling.

## 2 RELATED WORK

**Novel Class Discovery.** NCD is a relatively new problem proposed in recent years, aiming to discover novel classes (i.e., assign them to several clusters) by making use of similar but different known classes. The first two works that proposed NCD and tried to solve it were *KL-Divergence-based contrastive loss* (KCL) (Hsu et al., 2018) and meta classification likelihood (MCL) (Hsu et al., 2019), which employed feature extractors to predict pairwise similarity of each novel-class data pair. Han et al. (2019) proposed the *deep transfer clustering* (DTC) to first learn data embedding with metric learning on labeled data, then to employ the deep embedded clustering method (Xie et al., 2016) to cluster the novel-class data. To further extract information from data embedding, they proposed to use the *ranking statistics* (RS) to yield the pairwise similarity and use self-supervised learning to boost feature extraction (Han et al., 2020; 2021).

Recently, OpenMix (Zhong et al., 2021b) was proposed to mix known-class and novel-class data to learn a joint label distribution, benefiting to find their finer relations. Then a *neighborhood contrastive learning* (NCL) (Zhong et al., 2021a) was proposed to generate better discriminative representations. Fini et al. (2021) used pseudo-labels in combination with ground-truth labels in a *UNified Objective function* (UNO) that enabled better cooperation and less interference without self-supervised learning. Zhao & Han (2021) proposed a two branch method focused on local and global information, and used mutual knowledge distillation to promote information exchange and agreement.

Another work (Chi et al., 2022) demystify the assumption of NCD and formulated NCD with a sampling process. They argued that the novel classes and known classes should have similar high-level semantic meanings. Based on this assumption, they proved that NCD can be theoretical addressed and linked NCD to meta-learning that served as the solution in their paper.

**Label-Noise Learning.** Label-noise learning is to train an effective model with corrupted labels. Some works (Li et al., 2021b; Liu & Tao, 2016; Yao et al., 2020) estimated noise transition matrix to recover ground-truth labels. Based on the trick that small-loss data can be viewed as clean ones, Han et al. (2018) and Li et al. (2020) tried to filter clean data. Besides, Ren et al. (2018) used meta-learning on clean labeled data to boost sample weight and transition matrix.

**Deep Clustering.** Deep clustering aims to identify classes in an unsupervised way based on deep neural networks. Van Gansbeke et al. (2020) proposed to use self-supervised learning to obtain features as a priori in a learnable approach. Zhan et al. (2020) proposed an effective joint clustering and feature learning paradigm via decomposing feature clustering and integrating the process into iterations of network update. Yang et al. (2020) proposed a powerful adversarial attack algorithm to learn a small perturbation which can fool the clustering layers but not impact the deep embedding.

## 3 NCD UNDER UNRELIABLE SAMPLING

In this section, we first review the definition of NCD, and then formulate a new and more realistic problem called *NCD under unreliable sampling* (NUSA), and show that the existing NCD methods will fail to solve NUSA at last.

**Definition 1** (Novel Class Discovery). *In a sampling process, given a target label set $\mathcal{I}^{l}$ (i.e., known classes), we can collect known-class data $D_{\text{clean}}^{l} = \{(\boldsymbol{x}_i^l, y_i)\}_{i=1}^{N^l}$ and also unlabeled novel-class*

data $D_{\text{clean}}^{\text{u}} = \{\boldsymbol{x}_i^{\text{u}}\}_{i=1}^{N^{\text{u}}}$ with label set $\mathcal{I}^{\text{u}}$, where $y_i \in \mathcal{I}^{\text{l}}$, $\mathcal{I}^{\text{l}}$ and $\mathcal{I}^{\text{u}}$ contain $C^{\text{l}}$ and $C^{\text{u}}$ classes[2]. Moreover, $\mathcal{I}^{\text{l}}$ and $\mathcal{I}^{\text{u}}$ have similar high-level semantic meaning (*Chi et al., 2022*) but $\mathcal{I}^{\text{l}} \cap \mathcal{I}^{\text{u}} = \varnothing$. The aim of NCD is to learn a clustering model for novel classes using $D_{\text{clean}}^{\text{l}}$ and $D_{\text{clean}}^{\text{u}}$.

**Remark 2.** $C^u$ *is always assumed to be prior knowledge in NCD literature. In practice, however, if* $C^u$ *is unknown, an alternative is to use heuristic algorithms (e.g., Elbow method (Thorndike, 1953), Silhouette score (Rousseeuw, 1987)) to estimate it.*

**Sampling Errors.** In practice, sampling errors are common especially in professional fields, making NCD process unreliable. In this paper, we consider two important cases of sampling errors. One is misidentifying the known classes (i.e., the blue boxes in Figure 1(b)), named *internal* error:

**Definition 3** (Internal Sampling Error). *Given a known-class label set $\mathcal{I}^{\text{l}}$ and a collected known-class dataset $D^{\text{l}} = \{(\tilde{\boldsymbol{x}}_i^{\text{l}}, \tilde{y}_i)\}_{i=1}^{N^{\text{l}}}$, we say $D^{\text{l}}$ contains internal sampling errors if these is an $i_0$ such that $\tilde{y}_{i_0} \neq y_{i_0}$, where $y_{i_0} \in \mathcal{I}^{\text{l}}$ is the ground-truth label of $\tilde{\boldsymbol{x}}_{i_0}^{\text{l}}$.*

In addition, another case is confusing known classes and novel classes (i.e., the yellow boxes in Figure 1(b)), named *external* error:

**Definition 4** (External Sampling Error). *Given a known-class label set $\mathcal{I}^{\text{l}}$, a collected known-class dataset $D^{\text{l}} = \{(\tilde{\boldsymbol{x}}_i^{\text{l}}, \tilde{y}_i)\}_{i=1}^{N^{\text{l}}}$ and a collected novel-class dataset $D^{\text{u}} = \{\tilde{\boldsymbol{x}}_i^{\text{u}}\}_{i=1}^{N^{\text{u}}}$, we say there are external sampling errors between $D^{\text{l}}$ and $D^{\text{u}}$ if 1) there exists an $\tilde{\boldsymbol{x}}_i^{\text{l}}$ whose ground-truth label $y_i \in \mathcal{I}^u$ or 2) there exists an $\tilde{\boldsymbol{x}}_i^{\text{u}}$ whose ground-truth label $y_i \in \mathcal{I}^{\text{l}}$, where $\mathcal{I}^{\text{u}}$ is the novel-class label set that is unknown in advance.*

Note that, since known classes and novel classes have similar high-level semantic features, if collectors make mistakes when sampling known classes, they will probably confuse some specific known classes and novel classes. Namely, the above two types of sampling error often simultaneously occur when facing professional and difficult sampling tasks.

**Problem Setup of NUSA.** Based on both kinds of sampling errors, we can formulate a more realistic problem called *NCD under unreliable sampling* (NUSA) as follows.

**Definition 5** (NUSA). *Given $\mathcal{I}^{\text{l}}$ and $\mathcal{I}^{\text{u}}$ defined in Definition 1, in a sampling process, we can collect known-class data $D^{\text{l}} = \{(\tilde{\boldsymbol{x}}_i^{\text{l}}, \tilde{y}_i)\}_{i=1}^{N^{\text{l}}} \sim X^{\text{l}}$ and also unlabeled novel-class data $D^{\text{u}} = \{\tilde{\boldsymbol{x}}_i^{\text{u}}\}_{i=1}^{N^{\text{u}}} \sim X^{\text{u}}$, where $y_i \in \mathcal{I}^{\text{l}}$. The aim of NUSA is to learn a clustering model for novel classes by using $D^{\text{l}}$ and $D^{\text{u}}$ where $D^{\text{l}}$ contains internal sampling errors (Definition 3) and there are external sampling errors between $D^{\text{l}}$ and $D^{\text{u}}$ (Definition 4).*

**NUSA Degrades Existing NCD Methods.** As mentioned earlier, sampling errors may make NCD process unreliable. To verify this claim, we employ four existing NCD methods (i.e., DTC (Han et al., 2019), RS (Han et al., 2020), NCL (Zhong et al., 2021a), UNO (Fini et al., 2021)) to solve NUSA, shown in the *left* one of Figure 2. We find that the *clustering accuracy* (ACC) sharply drops as the sampling error rate (please refer to Section 5) increases from $10\%$ to $60\%$. Therefore, sampling errors negatively affect the performance of NCD methods. To ease the negative effects caused by sampling errors, we propose a *hidden-prototype-based discovery network* (HPDN), to resist sampling errors and keep good clustering performance in NUSA. We also give NUSA theoretical analysis and a learning upper bound in Appendix E.

## 4 HIDDEN-PROTOTYPE-BASED DISCOVERY NETWORK

The core challenge of NCD is to obtain good representations of novel-class data. However, for NUSA, these representations may be negatively affected by sampling errors. Thus, the keys to address NUSA are to obtain relatively clean representations and avoid overfitting to residual sampling errors.

To tackle the sampling errors and accurately separate novel-class data, we propose an effective framework HPDN (Figure 3), which is able to resist both *internal* error and *external* error mentioned in Definition 3 and 4. In Section 4.1, we train a deep network to fully fit the sampled data and try to yield clean data representations from hidden layers. In Section 4.2, we propose a mini-batch prototypical K-means algorithm, which further prevents clustering model overfitting to residual errors. Detailed method and its motivation are introduced in the following.

---

[2]$C^{\text{u}}$ is assumed to be prior knowledge (Zhong et al., 2021b)

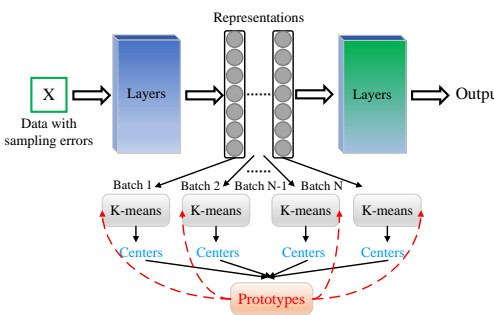

HPDN contains two modules. 1) Obtaining clean hidden representations of novel-class data (i.e., the *top* half of figure). After initialized by SimCLR (Chen et al., 2020), a deep network $f$ is firstly trained with reweighted known and novel class data, viewing all the novel classes as one class, i.e., class $C^l + 1$. Then, a proper hidden layer is used to yield the representations of novel-class data. 2) Clustering these representations with mini-batch prototypical K-means (i.e., the *bottom* half of figure). We divide a dataset into multiple mini-batches, and then cluster each batch by classical K-means. After obtaining the clustering centers of each mini-batch, we calculate their prototypes to serve as initial clustering centers of each mini-batch in the next epoch. Moreover, to avoids oscillating around the local minimum, current prototype memorizes the previous ones to control the learning rate.

Figure 3: Framework of *hidden-prototype-based discovery network* (HPDN).

## 4.1 ROBUST HIDDEN REPRESENTATION

Based on the discussion above, we will train a deep network to obtain relatively clean representations that are not seriously affected by sampling errors. Given $D^l$ and $D^u$ defined in Definition 5, we first initialize a deep network by SimCLR (Chen et al., 2020) without any supervision. We find that the sampled data with errors can be viewed as data with label noises from the perspective of supervision. According to (Li et al., 2021a), if an architecture "suits" one task, training with noisy labels can induce useful hidden representation. Thus, we temporarily view all the novel classes as the class $C^l + 1$ and generate $\hat{D}^u = \{(\boldsymbol{x}^u, C^l + 1) : \boldsymbol{x}^u \in D^u\}$. Then, we train a deep network $f : \mathcal{X} \to [0, 1]^{1 \times (C^l + 1)}$ to fully fit $D^l \cup \hat{D}^u$. As the data size of class $C^l + 1$ is more than others, we reweight each data according to the data amount of each class in standard cross-entropy loss to alleviate data imbalance, defined as,

$$\ell(\boldsymbol{x}_i, y_i; \theta_f) = -\Delta_{y_i} \boldsymbol{e}_{y_i} \log(f(\boldsymbol{x}_i)^T), \ \Delta_{y_i} = \frac{|D^l \cup \hat{D}^u|}{n_{y_i}(C^l + 1)}, \tag{1}$$

where $(\boldsymbol{x}_i, y_i) \in D^l \cup \hat{D}^u$, $\boldsymbol{e}_{y_i}$ denotes a $1 \times (C^l + 1)$ vector with a 1 in the $y_i$-th coordinate and 0's elsewhere and $\theta_f$ denotes the parameters of the deep network $f$. $n_{y_i}$ denotes the data amount of class $y_i$, and $|\cdot|$ denotes the number of elements in a set.

After the training procedure, $f$ can almost fully fit $D^l \cup \hat{D}^u$ (i.e., classification accuracy is more than 99% empirically), indicating that $f$ has overfitted the sampling errors. Based on the conclusion of (Li et al., 2021a) mentioned above, we try to employ proper hidden layers of $f$ to yield good representations for novel-class data. As a deep network, $f$ can be decomposed as $f = f_n \circ f_{n-1} \circ \cdots \circ f_1$, where $f_z$ denotes the $z$-th layer in the deep network $f$. Without loss of generality, we assume that the most clean representations are yielded by the $z$-th layer, i.e., $\psi_z(\boldsymbol{x}^u) := f_z(f_{z-1}(\cdots f_1(\boldsymbol{x}^u)))$, $\forall \boldsymbol{x}^u \in D^u$, $1 < z < n$. The choice of $z$ is discussed in Section 5. Although $\psi_z(\boldsymbol{x}^u)$ is enough good compared with $\psi_n(\boldsymbol{x}^u)$, it still contains residual sampling errors and causes continuous error accumulation in the training procedure of existing NCD methods (*right* one of Figure 2). To address this issue, we propose the mini-batch prototypical K-means, detaching the noisy supervision in time and dividing dataset into multiple batches for better initialization and less iterations.

## 4.2 MINI-BATCH PROTOTYPICAL K-MEANS

Existing NCD methods use various kinds of supervision to help cluster data, e.g., pairwise similarity (Han et al., 2020; Zhong et al., 2021a; Chi et al., 2022) and pseudo-label (Fini et al., 2021). These supervisions are obtained using data representations and thereby negatively affected by sampling errors for NUSA. Due to strong memory of deep networks (Zhang et al., 2021), the errors in supervision will continuously accumulate in the training procedure and invalidate existing NCD methods (Figure 2). Thus, we detach noisy supervision in time and employ fully unsupervised method, K-means (MacQueen et al., 1967). For naive K-means, all the data representations are

required at a time. We known K-means is very sensitive to initial centers (Arthur & Vassilvitskii, 2006). Selecting proper initial centers from all the representations contained residual errors is hard. Thus we propose the mini-batch prototypical K-means, which divides dataset into multiple batches to cluster respectively and takes the matched average centers (i.e., prototypes) as the initial centers of each batch in next epoch.

In detail, given unlabeled novel-class data $D^{\mathrm{u}}$ and batch size $B$, we partition $D^{\mathrm{u}}$ into $\lceil |D^{\mathrm{u}}|/B \rceil$ batches, i.e., $D^{\mathrm{u}} = D_1^{\mathrm{u}} \cup \cdots \cup D_{\lceil |D^{\mathrm{u}}|/B \rceil}^{\mathrm{u}}$, where $\lceil \cdot \rceil$ denotes the round up function. Through partition, mini-batch has smaller sample complexity so that is easier to initialize well (Canas et al., 2012). For each mini-batch, e.g., $\bar{D}_j^{\mathrm{u}}$, we firstly cluster $\bar{D}_j^{\mathrm{u}}$ by K-means, whose centers are initialized by K-means++ (Arthur & Vassilvitskii, 2006). Then, K-means output the clustering centers of $\bar{D}_j^{\mathrm{u}}$ (i.e., $\{c_i^{0,j}\}_{i=1}^{C^{\mathrm{u}}}$) and the assignments of each data. However, the clustering centers of all the mini-batches are very likely to be disordered, e.g., the first center in Batch A and the first center in Batch B do not represent the same category. Thus, we use the Hungarian algorithm (Kuhn, 1955) to align the centers of all the mini-batches and compute their prototype of the 0-th epoch defined as follows,

$$c_i^{0,*} = \frac{1}{\lceil |D^{\mathrm{u}}|/B \rceil} \sum_{j=1}^{\lceil |D^{\mathrm{u}}|/B \rceil} c_i^{0,j}, \quad i = 1, \ldots, C^{\mathrm{u}}. \tag{2}$$

**Remark 6.** *Note that there is an extreme case where there may be missing categories in some mini-batches. First of all, clustering data into all classes in a mini-batch will not make our algorithm crushed, but it indeed will introduce some errors in the optimization procedure. In our method, we shuffle novel-class data before dividing them in each updating step to alleviate this issue.*

The prototype $\{c_i^{0,*}\}_{i=1}^{C^{\mathrm{u}}}$ takes care of the entire $D^{\mathrm{u}}$ by memorizing the centers of each batch. To enforce each batch always consistent, we use the prototype $\{c_i^{0,*}\}_{i=1}^{C^{\mathrm{u}}}$ that we just obtained as the initial centers of each mini-batch in the next epoch. Through tuning, the variation of L2-norm of the prototype (i.e., the first term of equation 3) is very small empirically, indicating that the prototype converges to a stable state as the final clustering centers of novel-class data.

Moreover, to avoid oscillating around the local minimum in the iteration procedure, we let the current prototype memorize the previous prototypes to control the learning rate. In detail, when this algorithm enters into the epoch $t$, we obtain the prototype $\{c_i^{t,*}\}_{i=1}^{C^{\mathrm{u}}}$. For epoch $t+1$, we have

$$c_i^{t+1,*} = \frac{\beta}{\lceil |D^{\mathrm{u}}|/B \rceil} \sum_{j=1}^{\lceil |D^{\mathrm{u}}|/B \rceil} c_i^{t+1,j} + (1-\beta)c_i^{t,*}, \tag{3}$$

where $i = 1, \ldots, C^{\mathrm{u}}$ and $\beta$ is a hyper-parameter used to control the learning rate. Obviously, the larger $\beta$ indicates the larger learning rate. The choose of $\beta$ is detailed analyzed in Section 5. Therefore, the prototype of epoch $t+1$, $c_i^{t+1,*}$, memorizes the information of all the previous $t$ prototypes.

## 5 EXPERIMENTS

In this section, we conduct extensive experiments to verify the effectiveness of HPDN on NUSA, involving 3 benchmark datasets and 12 baselines.

**Datasets.** Following (Zhong et al., 2021a), we evaluate our method on three important benchmark datasets, including CIFAR-10 (Krizhevsky et al., 2009), CIFAR-100 (Krizhevsky et al., 2009), and ImageNet (Deng et al., 2009). We report the results averaged over 3 runs on CIFAR-10, CIFAR-100. For ImageNet, following (Han et al., 2020), we report the results averaged over 3 different label sets of novel-class data. The detailed strategy for partitioning known and novel classes is in Appendix A.

**Simulate sampling errors.** Since the datasets we choose are correct originally, we need to corrupt them manually to simulate the sampling errors through a transition matrix $Q \in [0,1]^{(C^{\mathrm{l}}+C^{\mathrm{u}}) \times (C^{\mathrm{l}}+C^{\mathrm{u}})}$, where $Q_{ij} = \mathbb{P}(\tilde{y} = j|y = i)$ is the probability that wrong label $\tilde{y}$ is flipped from ground-truth label $y$. Then, we give the precise definition of transition matrix:

Table 1: Experimental results on HPDN and other baselines. We report the ACC±standard deviation of ACC. All experiments are performed with sampling error rate of $40\%$ and cross rate of $50\%$. Bold values represent the highest average ACC in each column. We report the results averaged over 3 runs on CIFAR-{10,100}. For ImageNet, following (Han et al., 2020), we report the results averaged over 3 different label sets of novel-class data. Results of all the methods are trained for 100 epochs.

| Method | CIFAR-10 | CIFAR-100 | ImageNet | Average |
|---|---|---|---|---|
| Existing NCD methods | | | | |
| DTC (Han et al., 2019) | 28.51%±1.03% | 23.90%±0.77% | 27.32% | 26.58% |
| RS (Han et al., 2020) | 34.60%±0.62% | 21.28%±1.81% | 29.10% | 28.30% |
| NCL (Zhong et al., 2021a) | 33.76%±0.26% | 23.84%±0.65% | 33.06% | 30.22% |
| UNO (Fini et al., 2021) | 37.93%±2.47% | 26.11%±1.83% | 32.56% | 32.20% |
| Combine NCD methods with Co-teaching (Han et al., 2018) | | | | |
| DTC + Co-teaching | 46.28%±2.33% | 31.92%±0.79% | 38.76% | 38.97% |
| RS + Co-teaching | 46.17%±1.40% | 34.08%±2.10% | 36.31% | 38.85% |
| NCL + Co-teaching | 48.33%±0.57% | 35.74%±1.51% | 43.29% | 42.45% |
| UNO + Co-teaching | 52.18%±3.85% | 37.31%±3.14% | 46.13% | 45.21% |
| Combine NCD methods with DivideMix (Li et al., 2020) | | | | |
| DTC + DivideMix | 45.03%±1.97% | 33.74%±1.94% | 37.68% | 38.82% |
| RS + DivideMix | 47.52%±0.78% | 30.69%±2.07% | 40.44% | 39.55% |
| NCL + DivideMix | 51.56%±1.66% | 36.07%±0.75% | 46.51% | 44.71% |
| UNO + DivideMix | 50.10%±2.49% | 35.93%±1.88% | 46.27% | 44.10% |
| HPDN (Ours) | **63.18%±2.06%** | **37.96%±1.22%** | **53.60%** | **51.58%** |

where $\rho$ denotes the sampling error rate and $\tau$ denotes the cross rate. Specifically, given an instance to be sampled, $\rho$ represents the probability that its category is wrongly identified. Furthermore, given a fixed $\rho$, $\tau$ represents the probability that an known-class (*resp.* novel-class) instance is wrongly identified as a novel-class (*resp.* known-class) instance. Based on Definition 3 and 4, given the above transition

$$Q = \begin{bmatrix} 1-\rho & \frac{\rho(1-\tau)}{C^l-1} & \cdots & \frac{\rho\tau}{C^u} & \frac{\rho\tau}{C^u} \\ \frac{\rho(1-\tau)}{C^l-1} & 1-\rho & \cdots & \frac{\rho\tau}{C^u} & \frac{\rho\tau}{C^u} \\ \vdots & \vdots & \ddots & \vdots & \vdots \\ \frac{\rho\tau}{C^l} & \frac{\rho\tau}{C^l} & \cdots & 1-\rho & \frac{\rho(1-\tau)}{C^u-1} \\ \frac{\rho\tau}{C^l} & \frac{\rho\tau}{C^l} & \cdots & \frac{\rho(1-\tau)}{C^u-1} & 1-\rho \end{bmatrix},$$

matrix, the internal error rate is $\rho(1-\tau)$ and the external error rate is $\rho\tau$.

Note that, although the internal (*resp.* external) error rates are evenly assigned to known classes (*resp.* novel classes) in $Q$, this is not the only way to assign both errors between known classes and novel classes. Since this is the first work to consider such a hard problem, we would like to focus on this simple transition matrix at the current stage and leave more difficult transition matrices (e.g., instance-dependent transition matrices (Xia et al., 2020)) to future work.

**Baselines.** NUSA is a new problem and there is no straightforward solution to NUSA, thus we use related NCD methods and corresponding two-step methods as baselines. Related NCD methods include DTC (Han et al., 2019), RS (Han et al., 2020), NCL (Zhong et al., 2021a) and UNO (Fini et al., 2021). Simple reviews about these methods are in Section 2. Two-step methods are to sequentially combining label-noise learning methods and NCD methods. In detail, we firstly use label-noise learning methods, e.g., Co-teaching (Han et al., 2018) and DivideMix (Li et al., 2020), to correct the labels of known-class data and detect known (*resp.* novel) class data that are sampled as novel (*resp.* known) classes. Then we combine four NCD methods and two label-noise learning methods to form eight two-steps baselines whose abbreviations are shown in Table 5 (Appendix A).

**Evaluation metric.** For a clustering problem, we use the *average clustering accuracy* (ACC) to evaluate the performance of clustering, which is defined as $\max_{\phi \in L} \frac{1}{N} \sum_{i=1}^{N} \mathbb{1}\{\bar{y}_i = \phi(y_i)\}$, where $\bar{y}_i$ and $y_i$ denote the ground-truths and assigned indices. $L$ is the set of mappings from indices to ground-truths. We adopt the Hungarian algorithm (Kuhn, 1955) to find the optimal mapping and then obtain the final ACC with it. We also use more metrics (i.e., homogeneity, completeness, and v_measure) to evaluate NUSA methods in Appendix D. To evaluate how the number of novel classes affects the NUSA methods, we perform additional experiments shown in Appendix C.

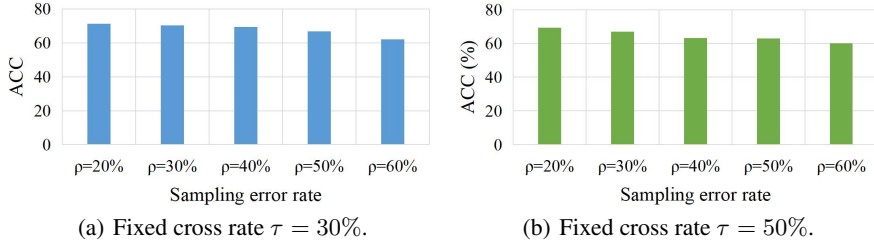

(a) Fixed cross rate $\tau = 30\%$.  (b) Fixed cross rate $\tau = 50\%$.

Figure 4: Experimental analysis about the robustness of HPDN under serious sampling errors. We take CIFAR-10 with the cross rates of 30% (a) and 50% (b) as an example. The sampling error rate and cross rate are introduced in Section 5. With the sampling error rate increasing from 20% to 60%, the lengths of corresponding columns are almost equal, indicating that HPDN is robust enough to serious sampling errors in NUSA.

**Implementation details.** The details about network structures and hyperparameters are in Appendix G.

**Comparison to baselines.** We compare HPDN with four existing NCD methods and eight two-step baselines as mentioned above. Both RS (Han et al., 2020) and NCL (Zhong et al., 2021a) need a model that is pretrained with self-supervised learning on all the data and then is finetuned by supervised learning on known-class data to output the features of novel-class data. To make a fair comparison, we use the self-supervised pretrained models provided by RS and NCL.

From Table 1, we find that HPDN significantly outperforms the baselines on all three datasets. For existing NCD methods, the ACCs tend to decrease at a later stage (Figure 2), because they are easy to overfit the sampling errors. From the results of two-step baselines, they substantially outperform NCD methods, indicating label-noise learning methods can effectively eliminate the negative effects of sampling errors. However, errors cannot be completely eliminated especially when error rate is large (e.g., $\geq 20\%$), and residual errors will accumulate in training procedure and further degrade NCD methods. Thus, the strategy of detaching noisy supervision timely to avoid overfitting in HPDN is really effective. Another important phenomenon is that the performance of HPDN on CIFAR-100 is not very good. This is because the data size of each class is small (i.e., 100 data per class) can they are too fine-grained, and HPDN cannot well address this hard situation. In addtion, we show and analyze the results of HPDN and eight baselines under lower error rate (i.e., 20%) in Appendix B.

**Ablation study.** In this subsection, we evaluate the effectiveness of each major component of HPDN in Table 2. If we replace the hidden layer (i.e., the fourth block of ResNet-18 in our work) with the last layer to yield representations, the ACC will drop more than 15% and 20% under sampling error rates of 20% and 40% respectively. This demonstrates that the hidden layers are not affected too much by sampling errors (Li et al., 2021a). If we eliminate the process of tuning prototypes, HPDN will degrade into the naive K-means, causing the low ACC (drops more than 10%) with serious oscillation. If we eliminate $\beta$ in equation 3 (i.e., set $\beta$ to 1), the prototypes will oscillate around the optimal solution for many iterations, thus fail to converge to it.

**Verify the robustness of HPDN under serious errors.** In this subsection, we would like to verify the robustness of HPDN under data with serious sampling errors. Therefore, we show a histogram to compare the ACCs under the sampling errors rate of $\{20\%, 30\%, 40\%, 50\%, 60\%\}$ and the cross rate of $\{30\%, 50\%\}$, taking CIFAR-10 as an example.

Comparing Figure 4 with Figure 2, as the sampling error rate increases, the performance degradation of HPDN is negligible compared with other baselines. Thus, these experimental results verify the robustness of HPDN under serious sampling errors in NUSA.

**Analysis about the choice of $\beta$.** In equation 3, we use the hyper-parameter $\beta$ to control the learning rate of prototypes. To make its updating stabler and faster, we empirically explore the choices of initial $\beta$ (Figure 5). We choose the initial $\beta$ as $\{0.01, 0.05, 0.10, 0.15, 0.20, 0.25, 0.30, 0.50\}$ to observe the change of ACC. We find that larger $\beta$ may lead HPDN to be unstable and cause the performance degradation. Too small $\beta$ (e.g., 0.01) will make the convergence speed of HPDN too slow, requiring more than 100 iterations, and make HPDN hard to be optimal. Thus, we choose 0.05 as the initial value based on above empirical evaluation.

**Impact of batch sizes on mini-batch prototypical K-means.** For clustering stage of HPDN, we divide dataset into multiple mini-batches. Taking CIFAR-10 as an example, we evaluate

Table 2: Ablation study of HPDN, taking CIFAR-10 as an example.

| Method | $\rho = 0.2$ | $\rho = 0.4$ |
|---|---|---|
| HPDN w/o HR | 51.06%±3.20% | 42.25%±2.94% |
| HPDN w/o Pro | 58.05%±0.73% | 57.73%±1.05% |
| HPDN w/o $\beta$ | 60.71%±3.62% | 53.90%±1.81% |
| HPDN w/o SSL | 61.71%±0.77% | 56.46%±0.59% |
| HPDN | **69.42%±1.93%** | **63.18%±2.06%** |

We choose the sampling error rates of 20% and 40% respectively, with the cross rate of 50% fixed. **HR**: hidden representation, **Pro**: mini-batch prototypical K-means, $\beta$: the hyperparameter used to control learning rate in equation 3, **SSL**: self-supervised learning initialization. Bold values represent the highest average ACC in each column.

Table 3: Results of HPDN under different batch sizes, taking CIFAR-10 and $\rho = 40\%$ as an example.

| Batch size | 64 | 128 | 256 | 512 | 1024 | all |
|---|---|---|---|---|---|---|
| HPDN | 63.24%±2.42% | 63.18%±3.03% | 60.42%±1.73% | 64.78%±2.51 | 68.57%±1.81% | 57.33% |

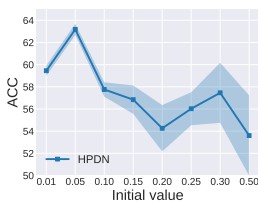

(a) Initial value of $\beta$.

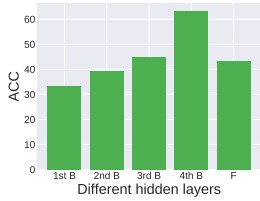

(b) Different hidden layers.

Take CIFAR-10 with $\rho = 40\%$ and $\tau = 30\%$ as an example. (a) HPDN is a little sensitive to initial. Smaller $\beta$ is more stable. $\beta$ (b) The fourth block of ResNet can yield best representations. "B" denotes the block and "F" denotes the fully-connected layer.

Figure 5: Analysis about the **initial value** of $\beta$ and **hidden layers**.

how batch size impacts the clustering performance empirically. We choose the batch size as $\{64, 128, 256, 512, 1024\}$ to observe the change of ACC. In addition, we use all the data representations at once to perform clustering stage as a comparison. Results is shown in Table 3. Firsts, we find mini-batch prototypical K-means substantially improves the clustering performance. Then, the ACCs under different batch sizes change little (almost within standard deviation) so that this indicates that batch size has little impact on HPDN. However, HPDN under smaller batch size requires more iterations, thus the convergence speed is slower. Moreover, HPDN under larger batch size requires more GPU memory and too large batch size will cause performance decreasing. To trade off, we choose 128 as the batch size in this work.

**Impact of different hidden layers on HPDN.** HPDN uses data representations yielded by hidden layers of deep networks, which are relatively clean compared with the representations of final layer. However, the quality of representations yielded by different hidden layers is also very different. We choose the representations yielded by 5 types of hidden layers (i.e., four blocks of ResNet and the following fully-connected layer) to perform clustering, and the results are shown in Figure 5. Obviously, the 4th block of ResNet can yield the best representations, which is the same as Li et al. (2021a). The $1 \sim 3$ blocks of ResNet cannot learn effective high-level features that are useful to clustering and the last fully-connected layer is seriously degraded by sampling errors.

## 6 CONCLUSION

Considering that sampling errors are common in real scenario of *novel class discovery* (NCD), this paper introduces a more realistic and more challenging problem: *NCD under unreliable sampling* (NUSA). However, existing NCD methods cannot handle NUSA well as the errors. To address this novel problem, we propose an effective method called *hidden-prototype-based discovery network* (HPDN). HPDN contains two modules: one is to obtain clean hidden-layer representations for novel-class data and another is to alternately cluster each mini-batches then aggregate them, detaching noisy supervision in time. We compare HPDN with 4 representative NCD methods and 8 competitive baselines on three benchmark datasets (CIFAR-10, CIFAR-100 and ImageNet). Empirical results demonstrate that HPDN can find better clustering centers for novel-class data compared to the 12 baselines. Especially, HPDN is robust to sampling errors and still performs well when facing serious sampling errors, which enables a new road to discover novel classes in some professional fields.

## 7 REPRODUCIBILITY STATEMENT

In this section, we briefly introduce how to reproduce our algorithm by yourself.

Our experiments are performed on Python 3.6.13, PyTorch 1.7.1, CUDA 11.2, and Tesla A100 GPUs.

The datasets that we use in this paper are all obtained from their official websites.

The main framework of our algorithm can be implemented according to the Alg. 1 with PyTorch.

The implementation details can be found in the following.

**Obtain Hidden Representations.** For a fair comparison with existing methods, we employ the ResNet-18/ResNet-50 (He et al., 2016) as the backbones of {CIFAR-10,CIFAR-100}/ImageNet. The backbone is initialized with SimCLR (Chen et al., 2020) for 300 epochs with the same training strategy as (Chen et al., 2020). Known-class data and novel-class data are randomly sampled from $D^{\mathrm{l}}$ and $D^{\mathrm{u}}$, whose batch size is set to 256/1024 for {CIFAR-10, CIFAR-100}/ImageNet. We use SGD optimizer with initial learning rate $0.1$, momentum $0.9$, and weigh decay $1e - 4$. In addition, the learning rate decays 10 times after each 40 epochs. We pretrain the backbone for 100/150 epochs for {CIFAR-10, CIFAR-100}/ImageNet. Then, we choose the outputs of the fourth block of ResNet with average pooling as the hidden representations.

**Mini-batch prototypical K-means.** The batch size is set to $128$ for all three datasets. We perform the clustering step for 100 epochs for all three datasets. For hyper-parameter $\beta$ in equation 3, it is initialized by $0.05$ and set to $0.05 * 0.5^{\mathrm{epoch}//20}$ in the training procedure, where "$//$" denotes the exactly divisible operation. We will further analyze the choice of $\beta$ in Section 5.

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

---

**Algorithm 1** Hidden-prototype-based Discovery Network (HPDN)

---

**Input**: deep network $f = f_n \circ f_{n-1} \circ \cdots \circ f_1$, known-class data $D^{\mathrm{l}} = \{(\boldsymbol{x}_i^{\mathrm{l}}, \tilde{y}_i)\}_{i=1}^{N^{\mathrm{l}}}$, novel-class data $D^{\mathrm{u}} = \{\boldsymbol{x}_i^{\mathrm{u}}\}_{i=1}^{N^{\mathrm{u}}}$, learning rate $\gamma$, batch size $B$, network parameters $\theta_f$, $z$ that using $f_z$ to extract representations, the maximum number of epochs $T$, hyper-parameter $\beta$;
**1: Initialize** $\theta_f$ and $t = 0$;
**2: Label** all the $\boldsymbol{x}^{\mathrm{u}} \in D^{\mathrm{u}}$ as the class $C^{\mathrm{l}} + 1$ and generate $\hat{D}^{\mathrm{u}} = \{(\boldsymbol{x}^{\mathrm{u}}, C^{\mathrm{l}} + 1) : \boldsymbol{x}^{\mathrm{u}} \in D^{\mathrm{u}}\}$;
*#phase one: extract robust hidden representations.*
**while** $t < T$ **do**
    **for** *each mini-batch* $\bar{D} \subset D^{\mathrm{l}} \cup \hat{D}^{\mathrm{u}}$ **do**
        **3: Compute** $L(\bar{D}; \theta_f) = \frac{1}{B} \sum_{(\boldsymbol{x},y) \in \bar{D}} \ell(\boldsymbol{x}, y; \theta_f)$ according to equation 1;  % Compute the average loss
        **4: Update** $\theta_f = \theta_f - \gamma \nabla_{\theta_f} L(\bar{D}; \theta_f)$;                % Update parameters of $f$
    **end**
**end**
*#phase two: mini-batch prototypical K-means.*
**while** $t < T$ **do**
    **for** *each mini-batch* $\bar{D}_j^{\mathrm{u}} \subset f_z(D^{\mathrm{u}}) := \{f_z(\boldsymbol{x}^{\mathrm{u}}) : \boldsymbol{x}^{\mathrm{u}} \in D^{\mathrm{u}}\}$ **do**
        **5: Initialize** clustering centers $\{c_i^{t,j}\}_{i=1}^{C^{\mathrm{u}}}$ using K-means++ **if** $t = 0$; **otherwise** using $\{c_i^{t,*}\}_{i=1}^{C^{\mathrm{u}}}$;
        **6: Cluster** $\bar{D}_j^{\mathrm{u}}$ with K-means algorithm and update clustering centers $\{c_i^{t+1,j}\}_{i=1}^{C^{\mathrm{u}}}$;
    **end**
    **8: Update** $c_i^{t+1,*}$ according to equation 3 and set $t = t + 1$;       % Update prototypes and $t$
**end**
**Output**: clustering centers $\{c_i^{T,*}\}_{i=1}^{C^{\mathrm{u}}}$

---

## A  PARTITION WAY OF THREE BENCHMARKS

As a key assumption in NCD, known classes and novel classes have similar high-level semantic features (Chi et al., 2022). Therefore, following existing works (Hsu et al., 2019; Han et al., 2019; Zhao & Han, 2021), we partition a dataset into two parts according to classes, where one part serves as the known-class group and the other one serves as the novel-class group. It is worth noting that there are no overlaps between known classes and novel classes, and the number of novel classes is assumed to be prior knowledge. The detailed way of partition can be seen in Table 4.

The influences of partition on NUSA (and NCD) methods are mainly in two aspects: 1) How similar the semantic features of known and unknown classes are; 2) How fine-grained the novel classes are. For the first aspect, the known-class data and novel-class data should have similar semantic features. If not, this setting will become cross-domain NUSA (and NCD), which is beyond our current research topic. Specifically, for CIFAR-10, they are more similar, while for ImageNet, they are less similar. For the second aspect, the novel classes of CIFAR-10 are dog, frog, horse, horse, truck, which are relatively coarse-grained, and the novel classes of CIFAR-100 are bicycle, bus, motorcycle, pickup truck, train,maple, oak, palm, pine, willow, and etc., which are more fine-grained. Overall, our experimental setup takes various situations of NUSA (and NCD) into account.

In addition, we evaluate how the number of novel classes, an important factor in clustering problem, matters in NUSA. Therefore, we test the performances of HPDN and baseline methods on NUSA with different numbers of novel classes, taking CIFAR-100 as an example. In detail, we choose the number of known classes as 80 like before, and we choose the numbers of novel classes as {20,10,5}, respectively. They are the 81-100 classes, 81-90 classes, and 81-85 classes in CIFAR-100, respectively.

Table 4: Partition way of three datasets.

| Dataset | #Known class | #Novel class |
|---|---|---|
| CIFAR-10 | 5 | 5 |
| CIFAR-100 | 80 | 20 |
| ImageNet | 882 | 30 |

Table 5: Abbreviations of two-step baselines.

| | DTC | RS | NCL | UNO |
|---|---|---|---|---|
| Co-teaching | C+D | C+R | C+N | C+U |
| DivideMix | D+D | D+R | D+N | D+U |

Table 6: Evaluate the impact of different numbers of novel classes on NUSA. Taking CIFAR-100 as an example, we choose the numbers of novel classes as 5, 10, and 20, respectively. All experiments are performed with the sampling error rate of 40% and the cross rate of 50%. Bold values represent the highest average ACC in each column. We report the results averaged over 3 runs.

| Number of novel classes | 5 | 10 | 20 |
|---|---|---|---|
| Existing NCD methods | | | |
| DTC (Han et al., 2019) | 26.72%±0.59% | 24.75%±0.43% | 23.90%±0.77% |
| RS (Han et al., 2020) | 42.67%±1.05% | 23.04%±1.44% | 21.28%±1.81% |
| NCL (Zhong et al., 2021a) | 27.83%±2.74% | 32.70%±0.89% | 23.84%±0.65% |
| UNO (Fini et al., 2021) | 44.03%±2.25% | 33.59%±1.66% | 26.11%±1.83% |
| Combine NCD methods with Co-teaching (Han et al., 2018) | | | |
| DTC + Co-teaching | 42.64%±0.49% | 34.35%±1.16% | 31.92%±0.79% |
| RS + Co-teaching | 56.81%±2.35% | 39.51%±1.41% | 34.08%±2.10% |
| NCL + Co-teaching | 48.38%±1.97% | 40.35%±1.13% | 35.74%±1.51% |
| UNO + Co-teaching | 60.72%±2.50% | **46.26%±2.39%** | 37.31%±3.14% |
| Combine NCD methods with DivideMix (Li et al., 2020) | | | |
| DTC + DivideMix | 39.04%±0.85% | 33.31%±1.22% | 33.74%±1.94% |
| RS + DivideMix | 50.27%±0.73% | 35.18%±1.32% | 30.69%±2,07% |
| NCL + DivideMix | 45.69%±2.53% | 39.83%±2.37% | 36.07%±0.75% |
| UNO + DivideMix | 60.89%±1,74% | 42.09%±2.67% | 35.93%±1.88% |
| HPDN (Ours) | **64.40%±1.53%** | 45.14%±1.77% | **37.96%±1.22%** |

## B RESULTS OF HPDN UNDER LOWER ERROR RATE

In Table 8, we show the results of HPDN and baselines with noise rate 20% and cross rate of 50%. We can find that HPDN almost outperforms all the baselines on three datasets, indicating that HPDN still works effectively under lower sampling error rate. However, for CIFAR-100. The number of test data of each class in CIFAR-10 is relatively few (i.e., each class has 100 data). HPDN relies on good initialization of clustering centers. When few data meets sampling errors, the initialization task will be hard for HPDN. We will further optimize the initialization process of HPDN to make it more robust.

## C IMPACT OF THE NUMBER OF NOVEL CLASSES ON NUSA

In this section, we evaluate how the numbers of novel classes affect the NUSA methods on CIFAR-100. Note that the clustering accuracy (ACC) that is commonly used as the metric in NCD/NUSA and other clustering-related problems does not take the number of novel classes into consideration. Thus, we specifically design experiments to evaluate the effectiveness of HPDN with regard to different numbers of novel classes.

We choose the number of known classes as 80 like previous works (Han et al., 2019; 2020), and we choose the numbers of novel classes as 20, 10, and 5, respectively. We report the ACCs of HPDN and all the baselines with different numbers of novel classes on CIFAR-100 in Table 6. We can find that the number of novel classes is a crucial factor for the performances of NUSA methods. All of these methods perform better when the novel classes are fewer. We can also find that HPDN outperforms baseline methods almost for every number of novel classes, except for 10. However, their gap is still within the error range. This mainly results from the relatively good robustness of UNO (Fini et al., 2021).

## D EVALUATE NUSA METHODS WITH MORE METRICS

To evaluate the NUSA methods more comprehensively and accurately, we employ another three metrics, homogeneity, completeness, and v_measure (Rosenberg & Hirschberg, 2007), to evaluate the HPDN and the baseline methods. A clustering result satisfies homogeneity if all of its clusters contain only data points which are members of a single class. A clustering result satisfies completeness if all the data points that are members of a given class are elements of the same cluster. V-measure

Table 7: Evaluate NUSA methods with more metrics. Taking CIFAR-10 as an example, we use three clustering metrics that are based on normalized conditional entropy to measure the NUSA methods, i.e., homogeneity, completeness, and v_measure respectively. All experiments are performed with the sampling error rate of $40\%$ and the cross rate of $50\%$. Bold values represent the highest average ACC in each column.

| Metrics | Homogeneity | Completeness | V_measure |
|---|---|---|---|
| Existing NCD methods | | | |
| DTC (Han et al., 2019) | 0.0563 | 0.0613 | 0.0587 |
| RS (Han et al., 2020) | 0.0506 | 0.1597 | 0.0769 |
| NCL (Zhong et al., 2021a) | 0.1769 | 0.1785 | 0.1777 |
| UNO (Fini et al., 2021) | 0.1618 | 0.1653 | 0.1635 |
| Combine NCD methods with Co-teaching (Han et al., 2018) | | | |
| DTC + Co-teaching | 0.2479 | 0.2527 | 0.2503 |
| RS + Co-teaching | 0.3139 | 0.3392 | 0.3261 |
| NCL + Co-teaching | 0.3552 | 0.3807 | 0.3675 |
| UNO + Co-teaching | 0.3871 | 0.4006 | 0.3937 |
| Combine NCD methods with DivideMix (Li et al., 2020) | | | |
| DTC + DivideMix | 0.2619 | 0.2778 | 0.2696 |
| RS + DivideMix | 0.3406 | 0.3955 | 0.3660 |
| NCL + DivideMix | 0.3797 | 0.4026 | 0.3908 |
| UNO + DivideMix | 0.4133 | 0.4310 | 0.4220 |
| HPDN (Ours) | **0.4742** | **0.4859** | **0.4800** |

is the harmonic average of homogeneity and completeness. These three metrics are based on the normalized conditional entropy, which measure the clustering performance in a different view from ACC and are commonly used in clustering problems. Constrained by space, we only report the results of HPDN and the baseline methods regarding these three metrics on CIFAR-10, which are shown in Table 7.

We can easily find that our method consistently outperforms the baseline methods with regard to these three normalized conditional entropy-based metrics. For almost all the methods, the homogeneity is slightly smaller than the completeness, indicating that there exist two or more classes are assign to the same cluster except the wrongly assigned data points. Thus, existing methods and our HPDN need to further improve the ability of clustering more fine-grained novel classes.

## E  THEORETICAL ANALYSIS OF NUSA

At beginning, we recall the definitions of NCD and NUSA.

**Definition 7** (NCD). *In a sampling process, given a target label set $\mathcal{I}^l$ (i.e., known-class label set), we can collect known-class data $D_{\text{clean}}^l = \{(\boldsymbol{x}_i^l, y_i)\}_{i=1}^{N^l} \sim X^l$ and also unlabeled novel-class data $D_{\text{clean}}^u = \{\boldsymbol{x}_i^u\}_{i=1}^{N^u} \sim X^u$ with label set $\mathcal{I}^u$, where $y_i \in \mathcal{I}^l$, $\mathcal{I}^l$ and $\mathcal{I}^u$ contain $C^l$ and $C^u$ classes respectively. Moreover, $\mathcal{I}^l$ and $\mathcal{I}^u$ have similar high-level semantic meaning (Chi et al., 2022) but $\mathcal{I}^l \cap \mathcal{I}^u = \varnothing$. The aim of NCD is to learn a clustering model for novel classes using $D_{\text{clean}}^l$ and $D_{\text{clean}}^u$.*

For a more realistic scenario, collectors may make mistakes in sampling tasks especially for professional fields, named *NCD under unreliable sampling* (NUSA).

**Definition 8** (NUSA). *Given $\mathcal{I}^l$ and $\mathcal{I}^u$ defined in Definition 7, in a sampling process, we can collect known-class data $D^l = \{(\tilde{\boldsymbol{x}}_i^l, \tilde{y}_i)\}_{i=1}^{N^l} \sim \tilde{X}^l$ and also unlabeled novel-class data $D^u = \{\tilde{\boldsymbol{x}}_i^u\}_{i=1}^{N^u} \sim \tilde{X}^u$, where $y_i \in \mathcal{I}^l$. The aim of NUSA is to learn a clustering model for novel classes by using $D^l$ and $D^u$ where $D^l$ contains internal sampling errors (Definition 3) and there are external sampling errors between $D^l$ and $D^u$ (Definition 4).*

For general NCD problem, existing work (Chi et al., 2022) pointed that NCD can be theoretically solvable with two key conditions: (A) transformation set of $X^l$ (denoted as $\Pi^l$) and transformation set

of $X^{\mathrm{u}}$ (denoted as $\Pi^{\mathrm{u}}$) are good enough to make their high-level semantic features totally separable; (B) $\Pi^{\mathrm{l}} \cap \Pi^{\mathrm{u}} \neq \varnothing$. For NUSA, however, the sampling errors may mislead the training process of transformations, so that the corresponding high-level semantic features invalid as empirically verified in Figure 2. In detail, we denote $\mathbb{P}_{X^{\mathrm{l}}}$ and $\mathbb{P}_{X^{\mathrm{u}}}$ as the distributions of $X^{\mathrm{l}}$ and $X^{\mathrm{u}}$ respectively, and the distributions $\mathbb{P}_{\tilde{X}^{\mathrm{l}}}$ and $\tilde{P}_{\tilde{X}^{\mathrm{u}}}$ corrupted by sampling errors are defined as,

$$\mathbb{P}_{\tilde{X}^{\mathrm{l}}} = (1 - \delta_{\mathrm{l}})\mathbb{P}_{X^{\mathrm{l}}} + \delta_{\mathrm{l}}\mathbb{P}_{X^{\mathrm{u}}}, \ \mathbb{P}_{\tilde{X}^{\mathrm{u}}} = (1 - \delta_{\mathrm{u}})\mathbb{P}_{X^{\mathrm{u}}} + \delta_{\mathrm{u}}\mathbb{P}_{X^{\mathrm{l}}},$$

where $\delta_{\mathrm{l}}$ (*resp.* $\delta_{\mathrm{u}}$) indicates that a proportion $\delta_{\mathrm{l}}$ (*resp.* novel) class data are incorrectly sampled. Thus, given data sampled from $\tilde{\mathbb{P}}_{X^{\mathrm{l}}}$ and $\tilde{\mathbb{P}}_{X^{\mathrm{u}}}$, if the learned transformations sets $\Pi^{\mathrm{l}}$ and $\Pi^{\mathrm{u}}$ still satisfy condition (A), NUSA also can be theoretically solved.

For next analysis, we model the sampling errors with transition (van Rooyen & Williamson, 2018). Given sample spaces $\mathcal{X}$ and $\tilde{\mathcal{X}}$, a transition from $P_1 \in \mathbb{P}(\mathcal{X})$ to $P_2 \in \mathbb{P}(\tilde{\mathcal{X}})$ is a linear map $T : \mathbb{P}(\mathcal{X}) \to \mathbb{P}(\tilde{\mathcal{X}})$. If $\mathcal{X}$ and $\tilde{\mathcal{X}}$ are finite, transition $T$ is just a matrix. For NUSA, internal errors and external errors may appear at the same time, thus we can jointly model them through transition. Given random variables $\tilde{X}^{\mathrm{l}}$ and $\tilde{X}^{\mathrm{u}}$ with sampling errors, they can be represented as,

$$\mathbb{P}_{\tilde{X}^{\mathrm{l}}} = Q(\mathbb{P}_{X^{\mathrm{l}}}), \ \mathbb{P}_{\tilde{X}^{\mathrm{u}}} = Q(\mathbb{P}_{X^{\mathrm{u}}}),$$

where $Q$ is the transition from ground-truth distribution to actual distribution with sampling errors. $\mathbb{P}_{X^{\mathrm{l}}}$ (*resp.* $\mathbb{P}_{X^{\mathrm{u}}}$) represents the probability distribution of $X^{\mathrm{l}}$ (*resp.* $X^{\mathrm{u}}$). As our aim is to eliminate the negative effects of sampling errors, we hope the transition $Q$ could be invertible.

**Definition 9** (Reconstructible transition (van Rooyen & Williamson, 2018)). *A transition $T \in \mathbb{T}(\mathcal{X}_1, \mathcal{X}_2)$ is reconstructible if $T$ has a left inverse; that is there exists a transition $R \in \mathbb{T}(\mathcal{X}_2, \mathcal{X}_1)$ such that $R \circ T = \mathbb{1}_{\mathbb{P}(\mathcal{X}_1)}$, where $\mathbb{P}(\mathcal{X}_1)$ denotes the set of all distributions on sample space $\mathcal{X}_1$.*

Thus, the left inverse of $T$ is its reconstruction. For general case, we can always take the Moore-Penrose pseudo inverse of $T$, $R = (T^*T)^{-1}T^*$, as the reconstruction, where $T^*$ is the dual operator of $T$. If $T$ itself is invertible, the reconstruction of $T$ is $R = T^{-1}$.

Our aim is to learn good transformation set $\Pi^{\mathrm{l}}$ (*resp.* $\Pi^{\mathrm{u}}$) satisfying conditions (A) and (B) with $\tilde{X}^{\mathrm{l}}$ and $\tilde{X}^{\mathrm{u}}$, such that the transformations in $\Pi^{\mathrm{l}}$ (*resp.* $\Pi^{\mathrm{u}}$) can yield high-level semantic features of samples drawn from $X^{\mathrm{l}}$ (*resp.* $X^{\mathrm{u}}$) to be totally separable. In detail, given a proper loss function $\ell : Y \times \mathcal{F} \to \mathbb{R}$, and our objective is to find $f \in \mathcal{F}$ such that $f$ can minimize

$$\sup_{\tilde{D} \sim \mathbb{P}_{\tilde{X}^{\mathrm{l}}}, \mathbb{P}_{\tilde{X}^{\mathrm{u}}}} \mathbb{E}_{(x^{\mathrm{u}}, y^{\mathrm{u}}) \sim \mathbb{P}_{X^{\mathrm{u}}}} \ell(y^{\mathrm{u}}, f(\tilde{D})(x^{\mathrm{u}})), \tag{4}$$

where $\mathcal{F}$ is the hypothesis space, and $D^u$ is the novel-class data sampled from $\mathbb{P}_{X^{\mathrm{u}}}$. $f(\tilde{D})$ represents the hypothesis trained on $\tilde{D}$ (i.e., data with sampling errors), and is obtained with the following objective,

$$\arg\min_{f \in \mathcal{F}} \mathbb{E}_{(x, \tilde{y}) \sim \mathbb{P}_{\tilde{X}^{\mathrm{l}}}, \mathbb{P}_{\tilde{X}^{\mathrm{u}}}} \ell(\tilde{y}, f(x)). \tag{5}$$

However, as $\tilde{D}$ contains sampling errors, directly training $f$ on $\tilde{D}$ using standard loss function does not make sense. As mentioned in (van Rooyen & Williamson, 2018), we can use the corruption corrected loss to eliminate the negative effects of sampling errors. As the sample spaces are finite here, expectation of one random variable function on one distribution can be viewed as inner product of them in Hilbert space. By properties of adjoint operator and definition of reconstruction, we have

$$\mathbb{E}_{\mathbb{P}} f = <\mathbb{P}, f> = <R \circ T(\mathbb{P}), f> = <T(\mathbb{P}), R^*(f)> = \mathbb{E}_{T(\mathbb{P})} R^*(f),$$

where $< \cdot, \cdot >$ denotes the inner product.

**Theorem 10** (van Rooyen & Williamson (2018)). *For all reconstructible transition $T$ and loss function $\ell : D \times \mathcal{F} \to \mathbb{R}$, the corruption corrected loss $\ell_R : \tilde{D} \times \mathcal{F} \to \mathbb{R}$ is defined as,*

$$\ell_R(\cdot, f) = R^*(\ell(\cdot, f)), \ \forall f \in \mathcal{F}.$$

*Then for all distribution $\mathbb{P}$, we have*

$$\mathbb{E}_{D \sim \mathbb{P}} \ell(D, f) = \mathbb{E}_{\tilde{D} \sim T(\mathbb{P})} \ell_R(\tilde{D}, f), \ \forall f \in \mathcal{F}.$$

*Proof.* This theorem can be directly derived according to the above discussion. □

In detail, we define the class probability distribution of a data point $(x, \tilde{y})$ that is outputted by the last layer of a deep network as $\delta(x) \in \mathbb{R}^{C^{\mathrm{l}}}$, where $f(x) = \arg\max_i \delta_i(x)$ and $\tilde{y}$ is the corrupted supervision information. With the reconstructible transition $T$, there exists

$$\tilde{\delta}_i = \mathbb{P}(\tilde{Y} = i) = \sum_j \mathbb{P}(\tilde{Y} = i | Y = j)\mathbb{P}(Y = j) = \sum_j T_{ji}\delta_j = T_i^\top \cdot \delta.$$

Through the reconstructibility of $T$, we can directly derive $\delta = (T^\top)^{-1}\tilde{\delta}$, which links the noisy supervision and ground-truth in the view of data representations. This result is consistent with Theorem 10. Next, we equivalently define the loss function with regard to the class probability distribution, i.e., $\mathcal{L}(\delta(x), y) := \ell(f(x), y) = \ell(\arg\max_i \delta_i(x), y)$. Then we can derive the following theorem to show how to obtain clean data representations under noisy supervision.

**Theorem 11.** *Let $f^* = \arg\min_{f \in \mathcal{F}} \mathbb{E}_{(x,y)\sim p}[\ell(f(x), y)]$ with $f^* = \arg\max_i \delta_i^*$ and $\mathcal{L}$ is k-Lipschitz. For any $f = \arg\max_i \delta_i \in \mathcal{F}$ learned with noisy supervision, we have*

$$\mathcal{R}(f) \leq \mathcal{R}(f^*) + k \cdot \|(T^\top)^{-1}\|_2 \cdot \mathbb{E}_p \|\tilde{\delta}(x) - \tilde{\delta}^*(x)\|_2,$$

*where $\tilde{\delta}(x) = T^\top \delta(x)$ and $\tilde{\delta}^*(x) = T^\top \delta^*(x)$.*

*Proof.*
$$\mathcal{R}(f) - \mathcal{R}(f^*) = \mathbb{E}_p[\ell(f(x), y) - \ell(f^*(x), y)] = \mathbb{E}_p[\mathcal{L}(\delta(x), y) - \mathcal{L}(\delta^*(x), y)]$$
$$= \mathbb{E}_p[\mathcal{L}((T^\top)^{-1}\tilde{\delta}(x), y) - \mathcal{L}((T^\top)^{-1}\tilde{\delta}^*(x), y)]$$
$$\leq \mathbb{E}_p[k \cdot \|(T^\top)^{-1}\tilde{\delta}(x) - (T^\top)^{-1}\tilde{\delta}^*(x)\|_2]$$
$$= \mathbb{E}_p[k \cdot \|(T^\top)^{-1}(\tilde{\delta}(x) - \tilde{\delta}^*(x))\|_2].$$

Thus, we have

$$\mathcal{R}(f) - \mathcal{R}(f^*) = |\mathcal{R}(f) - \mathcal{R}(f^*)|$$
$$= |\mathbb{E}_p[k \cdot \|(T^\top)^{-1}(\tilde{\delta}(x) - \tilde{\delta}^*(x))\|_2]|$$
$$\leq \mathbb{E}_p|k \cdot \|(T^\top)^{-1}(\tilde{\delta}(x) - \tilde{\delta}^*(x))\|_2|$$
$$\leq \mathbb{E}_p k \cdot \|(T^\top)^{-1}\|_2 \cdot \|\tilde{\delta}(x) - \tilde{\delta}^*(x)\|_2$$
$$= k \cdot \|(T^\top)^{-1}\|_2 \cdot \mathbb{E}_p \|\tilde{\delta}(x) - \tilde{\delta}^*(x)\|_2.$$
□

Theorem 11 tells us that if the reconstructible transition $T$ is known, the regret risk of the model trained with noisy supervision is bounded by $k \cdot \|(T^\top)^{-1}\|_2 \cdot \mathbb{E}_p \|\tilde{\delta}(x) - \tilde{\delta}^*(x)\|_2$. This error bound indicates that if the model fits noisy data very well, i.e., the term $\mathbb{E}_p \|\tilde{\delta}(x) - \tilde{\delta}^*(x)\|_2$ is very small, and then the regret risk $\mathcal{R}(f) - \mathcal{R}(f^*)$ will also be very small. Thus, the reconstructible transition can help us to obtain clean data representations under noisy supervision.

Based on Theorem 10 and Theorem 11, we can change our learning objective from equation 5 to

$$\arg\min_{f \in \mathcal{F}} \mathbb{E}_{(x,\tilde{y})\sim \mathbb{P}_{\tilde{X}^{\mathrm{l}}}, \mathbb{P}_{\tilde{X}^{\mathrm{u}}}} \ell_R(\tilde{y}, f(x)). \tag{6}$$

In this work, the sampling errors that we consider are class-dependent and can be simulated with the following transition matrix,

$$Q = \begin{bmatrix} 1-\rho & \frac{\rho(1-\tau)}{C^{\mathrm{l}}-1} & \cdots & \frac{\rho\tau}{C^{\mathrm{u}}} & \frac{\rho\tau}{C^{\mathrm{u}}} \\ \frac{\rho(1-\tau)}{C^{\mathrm{l}}-1} & 1-\rho & \cdots & \frac{\rho\tau}{C^{\mathrm{u}}} & \frac{\rho\tau}{C^{\mathrm{u}}} \\ \vdots & \vdots & \ddots & \vdots & \vdots \\ \frac{\rho\tau}{C^{\mathrm{l}}} & \frac{\rho\tau}{C^{\mathrm{l}}} & \cdots & 1-\rho & \frac{\rho(1-\tau)}{C^{\mathrm{u}}-1} \\ \frac{\rho\tau}{C^{\mathrm{l}}} & \frac{\rho\tau}{C^{\mathrm{l}}} & \cdots & \frac{\rho(1-\tau)}{C^{\mathrm{u}}-1} & 1-\rho \end{bmatrix},$$

as introduced in Section 5 in detail. It is easy to verify the determinant of $Q$ is nonzero, indicating that $Q$ is invertible. That is to say, the transition in NUSA is reconstructible, and there exists $\ell_R = (Q^{-1})^*(\ell)$ as the corrected version of $\ell$.

According to above discussion and the PAC-Bayes bound (Zhang, 2006), we have the following bound of learning with sampling errors (van Rooyen & Williamson, 2018).

**Theorem 12.** *For reconstructible transition $T$, algorithms $f : D \to \mathcal{F}$, distributions $\mathbb{P}_{X^l}$, $\mathbb{P}_{X^u}$, $\mathbb{P}_{\tilde{X}^l} = T(\mathbb{P}_{X^l})$ and $\mathbb{P}_{\tilde{X}^u} = T(\mathbb{P}_{X^u})$ and bounded loss function $\ell$,*

$$\mathbb{E}_{(x,y)\sim\mathbb{P}_{X^l},\mathbb{P}_{X^u}}\mathbb{E}_{\tilde{D}\sim\mathbb{P}_{\tilde{X}^l},\mathbb{P}_{\tilde{X}^u}}\ell(y,f(\tilde{D})(x)) \leq \mathbb{E}_{\tilde{D}=\{(x,\tilde{y})\}\sim\mathbb{P}_{\tilde{X}^l},\mathbb{P}_{\tilde{X}^u}}\ell_R(\tilde{y},f(\tilde{D})(x))+\|\ell_R\|_\infty\sqrt{\frac{2\log(|\mathcal{F}|)}{n}},$$

*where $\|\cdot\|$ denotes the infinite norm.*

Motivated by Theorem 12, we can turn to learn with data with sampling errors $\tilde{D}$,

$$f^* = \arg\min_{f\in\mathcal{F}} \mathbb{E}_{(x,\tilde{y})\in\tilde{D}}\ell_R(\tilde{y},f(x)). \tag{7}$$

Thus, there exists

$$\mathbb{E}_{(x,\tilde{y})\in\tilde{D}}\ell_R(\tilde{y},f^*(x)) \leq \mathbb{E}_{(x,\tilde{y})\in\tilde{D}}\ell_R(\tilde{y},f(x)) = \mathbb{E}_{(x,y)\sim\mathbb{P}_{X^l},\mathbb{P}_{X^u}}\ell(y,f(x)), \forall f \in \mathcal{F}. \tag{8}$$

Then, we can modify Theorem 12 to the following version.

**Theorem 13.** *For reconstructible transition $T$, algorithms $f : D \to \mathcal{F}$, distributions $\mathbb{P}_{X^l}$, $\mathbb{P}_{X^u}$, $\mathbb{P}_{\tilde{X}^l} = T(\mathbb{P}_{X^l})$ and $\mathbb{P}_{\tilde{X}^u} = T(\mathbb{P}_{X^u})$ and bounded loss function $\ell$,*

$$\mathbb{E}_{(x,y)\sim\mathbb{P}_{X^l},\mathbb{P}_{X^u}}\mathbb{E}_{\tilde{D}\sim\mathbb{P}_{\tilde{X}^l},\mathbb{P}_{\tilde{X}^u}}\ell(y,f^*(\tilde{D})(x)) \leq \inf_{f\in\mathcal{F}}\mathbb{E}_{(x,y)\sim\mathbb{P}_{X^l},\mathbb{P}_{X^u}}\ell(y,f(x))+\|\ell_R\|_\infty\sqrt{\frac{2\log(|\mathcal{F}|)}{n}}.$$

*Proof.* This theorem can be directly derived from Theorem 10 and Theorem 12. $\square$

From Theorem 13, we can find the learning risk mainly depends on $\|\ell_R\|$, which is decided by the transition $T$.

For naive training strategy, we aim to minimize

$$\mathbb{E}_{(x,\tilde{y})\in\tilde{D}}\ell(\tilde{y},f(x)),$$

where $f = f_n \circ f_{n-1} \circ \cdots \circ f_1$. However, in our method, we use the representations yielded by hidden layers of deep networks. In this paper, we employ the second last layer and turn to minimize

$$\mathbb{E}_{(x,\tilde{y})\in\tilde{D}}\ell(\tilde{y},f_{n-1}\circ f_{n-2}\circ\cdots\circ f_1(x)).$$

That is, we employ $\ell(\tilde{y}, f_n^{-1} \circ f(x))$ to approximate $\ell_R(\tilde{y}, f(x))$. $f_n$ is likely to be not invertible, but we can use the Moore-Penrose pseudo left inverse of $f_n$ instead. In this view, the last layer $f_n$ serves as the approximation of the transition $T$ implicitly.

## F   COMPARISON OF HPDN AND SOTA STANDARD NCD METHODS

HPDN is specifically designed for NCD under unreliable sampling (NUSA). Encountering sampling errors, HPDN will be much more robust according to Table 1 and 8. However, HPDN cannot outperform SOTA standard NCD methods, e.g., (Zhao & Han, 2021) and (Zhong et al., 2021a). The main difference between the SOTA standard NCD methods and HPDN is the clustering procedure (the warm-up procedures are similar). For standard NCD methods, e.g., (Zhao & Han, 2021) and (Zhong et al., 2021a), their main framework of clustering is to use the data representations induced by deep networks to compute pairwise similarity and then obtain the pairwise pseudo-labels, and the clustering problem is reduced to binary classification problem (Hsu et al., 2019). With the strong fitting ability of a deep network, they can achieve good performance in standard NCD. However, encountering sampling errors (i.e., NUSA), deep networks are also easy to overfit these errors due to their strong fitting ability. In addition, these errors will accumulate more and more in the training procedure, causing performance degradation (Figure 2). For HPDN, to alleviate the bad effect of sampling errors, we detach all the supervision information in time and propose an Mini-batch Prototypical K-means to perform clustering. K-means is a fully unsupervised method. With useful data representations, our Mini-batch Prototypical K-means manages to avoid the accumulation of sampling errors. As its limited fitting ability, our method may not be able to outperform SOTA standard NCD methods.

Table 8: Experimental results on HPDN and other baselines. We report the ACC±standard deviation of ACC. All experiments are performed with sampling error rate of $20\%$ and cross rate of $50\%$. Bold values represent the highest average ACC in each column. We report the results averaged over 3 runs on CIFAR-10, CIFAR-100. For ImageNet, following (Han et al., 2020), we report the results averaged over 3 different label sets of novel-class data. Results of all the methods are trained for 100 epochs.

| Method | CIFAR-10 | CIFAR-100 | ImageNet | Average |
|---|---|---|---|---|
| Existing NCD methods | | | | |
| DTC (Han et al., 2019) | 30.26%±1.64% | 25.10%±1.61% | 34.19% | 29.85% |
| RS (Han et al., 2020) | 34.56%±1.80% | 21.93%±0.68% | 35.02% | 30.50% |
| NCL (Zhong et al., 2021a) | 34.71%±0.75% | 25.07%±2.34% | 34.18% | 31.32% |
| UNO (Fini et al., 2021) | 40.19%±1.88% | 27.18%±1.90% | 36.92% | 34.76% |
| Combine NCD methods with Co-teaching (Han et al., 2018) | | | | |
| DTC + Co-teaching | 58.41%±3.22% | 39.27%±1.37% | 50.38% | 49.35% |
| RS + Co-teaching | 55.63%±1.85% | 39.60%±3.57% | 53.23% | 49.49% |
| NCL + Co-teaching | 57.96%±0.82% | 40.38%±1.06% | 47.17% | 48.50% |
| UNO + Co-teaching | 62.37%±2.50% | 42.26%±2.85% | 55.16% | 53.26% |
| Combine NCD methods with DivideMix (Li et al., 2020) | | | | |
| DTC + DivideMix | 60.48%±2.14% | 41.23%±3.11% | 52.93% | 51.55% |
| RS + DivideMix | 64.17%±2.08% | **44.21%±3.64%** | 54.16% | 54.18% |
| NCL + DivideMix | 61.40%±0.68% | 38.39%±1.84% | 50.92% | 50.24% |
| UNO + DivideMix | 65.59%±2.37% | 42.13%±2.39% | 54.79% | 54.17% |
| HPDN (Ours) | **68.85%±3.11%** | 43.72%±1.04% | **60.16%** | **57.58%** |

## G  IMPLEMENTATION DETAILS

Our experiments are performed on Python 3.6.13, PyTorch 1.7.1, CUDA 11.2, and Tesla A100 GPUs.

**Obtain Hidden Representations.** For a fair comparison with existing methods, we employ the ResNet-18/ResNet-50 (He et al., 2016) as the backbones of {CIFAR-10,CIFAR-100}/ImageNet. The backbone is initialized with SimCLR (Chen et al., 2020) for 300 epochs with the same training strategy as (Chen et al., 2020). Known-class data and novel-class data are randomly sampled from $D^l$ and $D^u$, whose batch size is set to 256/1024 for {CIFAR-10, CIFAR-100}/ImageNet. We use SGD optimizer with initial learning rate 0.1, momentum 0.9, and weigh decay $1e-4$. In addition, the learning rate decays 10 times after each 40 epochs. We pretrain the backbone for 100/150 epochs for {CIFAR-10, CIFAR-100}/ImageNet. Then, we choose the outputs of the fourth block of ResNet with average pooling as the hidden representations.

**Mini-batch prototypical K-means.** The batch size is set to 128 for all three datasets. We perform the clustering step for 100 epochs for all three datasets. For hyper-parameter $\beta$ in equation 3, it is initialized by 0.05 and set to $0.05 * 0.5^{\text{epoch}//20}$ in the training procedure, where "$//$" denotes the exactly divisible operation. We will further analyze the choice of $\beta$ in Section 5.

