# OpenReview forum: "Novel Class Discovery under Unreliable Sampling"
_ICLR.cc/2023/Conference — Submitted to ICLR 2023_

### Official Review · Reviewer_6g6E · 2022-10-21

**Confidence:** 3
**Clarity, Quality, Novelty And Reproducibility:** This paper is clearly written and nov…
**Correctness:** 3
**Technical Novelty And Significance:** 3
**Empirical Novelty And Significance:** 3
**Recommendation:** 6

**Strength And Weaknesses:**

Strength:
- This work tackles a relatively new potential problem of NCD that has not yet received much attention.
- This paper is well written and easy to follow.
- The proposed method is novel and well motivated.

Weaknesses:
- Some terms and conclusions are not well explained. For instance, what is the residual sampling errors? Why fully fitting the sampled data is necessary to yield clean data representations from hidden layers?

**Summary Of The Paper:**

This paper aims at a new and interesting setting of novel class discovery (NCD), namely NCD under unreliable sampling (NUSA). In my view it is common NCD plus a special case of label noise. Specifically, this paper is motivated by a sampling view of NCD, and identidies two sampling errors one may face. To conquer the sampling errors, this paper takes inspiration from current label noise literature and proposes a two-stage learning paradigm by first fitting the wrongly sampled data and then discovering the novel classes through a novel mini-batch k-means algorithm. The proposed method is evaluated on several popular NCD benchmarks, which has demonstrated the superiority against the sampling erorrs in NCD.


**Summary Of The Review:**

A good submission with an intersesting scope and solid solution.

---

> ### Author Response · Authors · 2022-11-18
> **Responses to Reviewer 6g6E**
>
> Thanks for your constructive comments! We will address your major concerns in the following.
>
> >Q1. Some terms and conclusions are not well explained. For instance, what is the residual sampling errors?
>
> A1. Thanks for your carefulness. We will go through our paper, and try to find and improve the existing vague expressions.
>
> As discussed in our paper, although there exist noisy labels in the training data, the proper hidden layers of deep networks also can induce high-quality data representations. However, in practice, these noisy labels always negatively affect the data representations slightly. For our proposed setting, NUSA, the noisy labels come from the process of data sampling. Based on the above truths, the residual sampling errors are the errors embedded in the hidden representations, yielded by the sampling errors.
>
> >Q2. Why fully fitting the sampled data is necessary to yield clean data representations from hidden layers?
>
> A2. This conclusion is derived from the special properties of deep neural networks. In detail, previous works (e.g.,[1][2]) pointed out that neural networks tend to learn easy knowledge first, and label noise is mainly fitted by the very latter layers of deep neural networks. In addition, this conclusion is systematically studied and proposed by [3]. They perform solid experiments to verify that the hidden representations are still useful even the noise rate is beyond 50%, and give a corresponding theoretical analysis.
>
> For NUSA, there exist sampling errors between the known and novel classes and within them. Such sampling errors are able to be viewed as label noises equivalently in the framework of supervised learning. Thus, the data representations induced by hidden layers are significant for NUSA.
>
> Reference
>
> [1] Han et al. Co-teaching: Robust training of deep neural networks with extremely noisy labels. NeurIPS, 2018.
>
> [2] Wei et al. To smooth or not? when label smoothing meets noisy labels. ICML, 2022.
>
> [3] Li et al. How does a neural network’s architecture impact its robustness to noisy labels? NeurIPS, 2021.

---

### Official Review · Reviewer_SxkT · 2022-10-22

**Confidence:** 4
**Correctness:** 4
**Technical Novelty And Significance:** 3
**Empirical Novelty And Significance:** 2
**Recommendation:** 3

**Clarity, Quality, Novelty And Reproducibility:**

I think the work is phrased clearly in that I found it easy to follow. However, I believe that the method is phrased in too complicated a fashion, and in such a way that it obfuscates its simplicity and relation to prior work.

I would say the quality of this paper is medium, with a commendable number of baselines implemented and reasonable ablations given.

The originality of the work is also medium. The setting is an extension of an exisiting problem which has not been studied before, though the method bears strong resemblance to existing work. (I note that this latter point is not a problem in and of itself, though I find it problematic that it is phrased in a complicated manner).

**Strength And Weaknesses:**

Strengths:
* The problem the authors tackle is interesting. Label noise is something practical vision systems must deal with and the authors demonstrate that the performance of existing methods deteriorates as noise is introduced. Furthermore, the authors explore two forms of label-noise, within the closed-set classes and between the closed and open-set categories.
* The authors have made a good effort to create strong baselines by running on the combinatorial space of four NCD baselines and two label-noise cleaning techniques. However, it is not clear if and how hyper-parameters for these methods were tuned.
* The authors' proposed solution outperforms all the baselines. Furthermore, they validate all design choices in the ablation, with substantial performance drops without each component.

Weaknesses:
* My main concern is that I am not wholly convinced by the method. Though the authors phrase it in quite a complicated way ('Hidden-prototype-based discovery network'), the method is actually very simple: train a model with cross-entropy loss on all data (treating all unlabelled instances as one class); and then cluster network features with mini-batch K-Means. The language used to phrase the method seems more confusing than explanatory. For instance, despite the discussion of how earlier features may have fit less to label noise, the authors seem to use features before the final linear layer, which is a standard choice for clustering of deep features. This is especially a problem since in some cases, the improvements over baselines are small (e.g CIFAR100 case).
    * I would strongly suggest the authors re-phrase the method section to highlight the simplicity of the method and its strong relation to prior work (e.g MiniBatchK-Means implemented in sklearn).
* The authors motivate this study by stating that experts or professionals often confuse similar object categories. However, all results are shown on coarse-grained datasets, where the categories (e.g 'horse' and 'ship' in CIFAR10) are easy to distinguish. Other category discovery papers [1] have evaluated on fine-grained datasets which better reflect the motivating setting.
* Fig1b shows that the performance of RankStats alone can reach as high as 41%, though the performance in Table 1 is reported at 34%. Is this because performance at the final checkpoint is taken? If so, there should be ways to perform early stopping (like performance on a validation set of closed-set examples).

[1] Generalized Category Discovery, Vaze et al., CVPR22


**Summary Of The Paper:**

This paper tackles the problem of novel category discover (NCD) in the setting where samples have been mis-labelled. Standard NCD gives a model a labelled set of images from closed-set categories, as well as an unlabelled set of images from disjoint categories. The task becomes to learn a classifier which can both recognise the labelled classes but also cluster new categories in the unseen data. In this paper, the authors consider the setting in which there is noise within closed-set categories (the labelled set), but also when some closed-set examples appear in the unlabelled set and vice-versa. They name this setting 'NUSA'.

The authors develop a method to solve this problem which involves fitting a standard classifier on all of the data with a (weighted) cross-entropy loss, treating all unlabelled samples as a single class. Clustering of the new classes is performed by running mini-batch K-Means, where centroids are identified on mini-batches and then averaged across them all.

The authors show results on CIFAR10, CIFAR100 and ImageNet, which are standard NCD benchmarks. The authors create baselines by taking standard NCD methods and running them on top of data which has been algorithmically cleaned with methods from the label-noise literature.

**Summary Of The Review:**

In its current form, I recommend rejecting the paper. I believe this paper makes promising initial steps towards an interesting problem, but I believe both the methods and evaluations could be expanded before it is accepted as a conference publication.

UPDATE AFTER AUTHOR RESPONSE:

I have now read the other reviews and the authors' corresponding comments. I first acknowledge the authors' positive response to my suggestion for early stopping to improve the performance of the baselines. I am surprised by the authors' finding that early stopping actually reduces the performance of the baseline further (though I would be interested in seeing the training curves for validation accuracy vs overall accuracy to verify this).

Overall, however, my main concerns behind the paper remain. Specifically, though the setting is interesting, the method is essentially to train a supervised representation by treating all unlabelled instances as one class, followed by (mini-batch) k-means. This is described in an unnecessarily complicated way which obfuscates the simplicity of the method, and the authors have not updated the paper to reflect this. The result is that I find it difficult to understand which technical components lead to improvement over the baselines.

For instance, lengthy discussions on the robustness of the intermediate representations (final paragraph of 4.1) are unnecessary given the authors eventually run the classifier on top of the networks' penultimate layer (Fig 5b). This choice is entirely standard (it is the 'feature vector', or the representation before the FC classifier). I also still fail to see substantial difference between the authors proposed mini-batch k-means and the standard version in sklearn, though perhaps I am missing something.

As such, though I believe the direction of research is interesting, I maintain my original rating.

---

> ### Author Response · Authors · 2022-11-18
> **Responses to Reviewer SxkT - Part 2**
>
> >Q2. The authors motivate this study by stating that experts or professionals often confuse similar object categories. However, all results are shown on coarse-grained datasets, where the categories (e.g 'horse' and 'ship' in CIFAR10) are easy to distinguish. Other category discovery papers [1] have evaluated on fine-grained datasets which better reflect the motivating setting.
>
> A2. Thanks for your kind reminder. Fine-grained datasets are indeed more practical for the setting of NUSA. CIFAR-100 and ImageNet that we have evaluated are fine-grained to some degree. As we have discussed in Appendix A, for instance, CIFAR-100 contains the categories of bicycle, motorcycle, bus, pickup truck, etc. These categories are fine-grained enough to evaluate the effectiveness of our method.
>
> According to your advice, we find that [1] proposed to use the Semantic Shift Benchmark (SSB, including CUB and Stanford Cars) to evaluate their NCD method. CUB is about the birds and Stanford Cars is about the cars. They are specialized fine-grained visual recognition datasets. We will add the experimental results of these two datasets to the revised version once they have been finished.
>
> >Q3. Fig1b shows that the performance of RankStats alone can reach as high as 41%, though the performance in Table 1 is reported at 34%. Is this because performance at the final checkpoint is taken? If so, there should be ways to perform early stopping (like performance on a validation set of closed-set examples).
>
> A3. Yes, the 41% in Figure 2b is the best intermediate result, while the 34% in Table 1 is the final reported result. This phenomenon exactly indicates that deep learning-based clustering methods inevitably bring about serious overfitting. As you suggested, early-stopping is an effective trick to alleviate overfitting. However, there exists an issue that we do not have clean validation set to decide when to stop in NUSA. In detail, the known-class set does not have ground-truth labels and the novel-class set is unlabeled, and these two sets are also confusable.
>
> However, we still try to apply early-stopping into the supervised training stage of the existing NCD methods, and the corresponding final results are shown in the tables below, taking CIFAR-10 as an example. We randomly split 20% of the known-class set as the validation set, and use the loss values on the validation set as the criterion. The patience of early-stopping is set to be 7.
>
> Sampling error rate=40%, ES represents the early-stopping
> |        | DTC | RS | NCL |
> |--------|-----|----|-----|
> | w/o ES |28.51%|34.60%|33.76%|
> | + ES   |30.21%|27.78%|32.18%|
>
>
> Sampling error rate=20%, ES represents the early-stopping
> |        | DTC | RS | NCL |
> |--------|-----|----|-----|
> | w/o ES |30.26%|34.56%|34.71%|
> | + ES   |35.02%|28.32%|40.04%|
>
> From the above results, we find that early-stopping is not suitable for NUSA, as the clean labeled validation set is inaccessible. We cannot obtain consistent improvements over the original results from this trick. In addition, the noisy validation data even negatively affect the existing model performance. However, this still points us in a possible direction, and we will continue to explore it in the future.
>
> Reference
>
> [1] Vaze et al. Generalized Category Discovery, CVPR 2022.
>
> [2] Li et al. How does a neural network’s architecture impact its robustness to noisy labels? NeurIPS, 2021.

---

> ### Author Response · Authors · 2022-11-18
> **Responses to Reviewer SxkT - Part 1**
>
> Thanks for your constructive comments! We will address your major concerns in the following.
>
> >Q1. My main concern is that I am not wholly convinced by the method. Though the authors phrase it in quite a complicated way ('Hidden-prototype-based discovery network'), the method is actually very simple.
>
> A1. We now realize that the phrase of our method (i.e., hidden-prototype-based discovery network) is a little complicated, and our original purpose is to explicitly show most of the key elements of our method. As you point out, such a phrase may be not very friendly for readers. Therefore, we come up with another more elegant one, called "**er**ror-fr**i**endly novel **c**l**a**ss discovery (Erica)". We will consider changing the method name to this one.
>
> **However**, we have to emphasize that our method is **not** naively composed of training a classification model and mini-batch K-means. There exist clear motivations and original techniques within it. The detailed explanations are presented below.
> - Inspired by the conclusion of [2], we find that training a classifier to fully fit the noisy data is a simple yet effective strategy to obtain relatively clean data representations for NCD with sampling errors, requiring no additional complicated techniques and auxiliary data.
> - Motivated by the empirical results shown in Figure 2b, we find that the existing NCD methods with deep clustering seriously overfit to noisy supervision (e.g., pairwise similarity and pseudo label) in the training procedures, bringing about the corresponding raise&fall curves. This is because of the powerful memory of deep neural networks. Thus, we detach the noisy supervision in time and design a fully unsupervised clustering method, i.e., mini-batch prototypical K-means.
> - Our proposed mini-batch prototypical K-means is different from the mini-batch K-means implemented in Sklearn. The mini-batch K-means in Sklearn: 1) it uses the first batch to train a K-means model 2) it uses the current clustering centers as the initialization of the second batch, and so on. This method does not consider all the data at once, and will bring more and more biases in the updating process. The official document of sklearn also claims that this mini-batch K-means may underperform naive K-means.
> However, our proposed mini-batch prototypical K-means alleviates the above issue. We first cluster each batch, respectively, and calculate the prototypes of clustering centers in every batch (after aligning the clustering centers of each batch by the Hungarian algorithm) as the initial centers of the next iteration. Thus, our method considers all the data at once under the mini-batch condition, effectively alleviating the biases mentioned above.

---

### Official Review · Reviewer_g5Zb · 2022-10-25

**Confidence:** 4
**Correctness:** 2
**Technical Novelty And Significance:** 2
**Empirical Novelty And Significance:** 1
**Recommendation:** 3

**Clarity, Quality, Novelty And Reproducibility:**

Overall, the paper is fairly clearly written, barring a few grammatical or typographical errors.
The novelty of the paper is hard to appreciate given the lack of justification of the baselines and the choice of performance metrics.
The authors have included the details for reproducing the paper.

**Strength And Weaknesses:**

Strengths:
The authors have clearly stated the problem they are addressing and discussed a solution along with experimental results. They have demonstrated the robustness of HPDN under various conditions.

Weakness:
The results of the HPDN model are significantly better than the other approaches, which makes one wonder if (a) the baselines chosen are weak, or (b) if the model is indeed better. It is not clear from the paper which is correct.
Looks like approaches such as UNO (https://arxiv.org/pdf/2108.08536.pdf) and AutoNovel (https://arxiv.org/pdf/2106.15252v1.pdf) seem to have reported different performance metrics than those reported in this paper.

Further, it is not clear why k-means was chosen instead of other approaches such as cosine similarity, nearest neighbors etc.

**Summary Of The Paper:**

The authors describe an approach to solve the unreliability in the novel class discovery problem (NCD), where collectors may misidentify known classes and even confuse known classes with novel classes. They propose a solution which is a combination of a proposed deep network called hidden-prototype-based discovery network (HPDN), and mini batch k-means. The authors evaluate their approach against 3 benchmark datasets and 3 types of models (Existing NCD methods, Combine NCD methods with Co-teaching and Combine NCD methods with DivideMix).

**Summary Of The Review:**

Overall, the paper addresses an important problem of handling novel class discovery problem (NCD). The authors have proposed an approach that learns a HPDN and uses k-means for clustering. The metrics are surprisingly better than baselines, but also are different from those reported in the papers.

---

> ### Author Response · Authors · 2022-11-18
> **Responses to Reviewer g5Zb**
>
> Thanks for your constructive comments! We will address your major concerns in the following.
>
> >Q1. The results of the HPDN model are significantly better than the other approaches, which makes one wonder if (a) the baselines chosen are weak, or (b) if the model is indeed better. It is not clear from the paper which is correct. Looks like approaches such as UNO  and AutoNovel seem to have reported different performance metrics than those reported in this paper.
>
> A1. The chosen baselines of this paper are constituted from four NCD methods (including two representative methods (DTC, RS) and two SOTA methods (NCL, UNO)) and together with two competitive label-noise learning methods (Co-teaching, DivideMix), forming 12 baselines in total.
>
> First, NUSA is an important setting extended from NCD, and there is no specific method at the moment. Thus, the SOTA NCD methods are necessary to serve as the basic baselines. In addition, NUSA considers the scenario where there exist errors in the data collecting process of NCD. These sampling errors are able to be viewed as label noises equivalently in the framework of supervised learning. Thus, we first choose two current competitive label-noise learning methods to correct the sampling errors, and apply the corrected data to the four NCD methods mentioned above. We think our chosen baselines are competitive enough at the current stage.
>
> As for the performance metric, we carefully check the papers of UNO and AutoNovel again, and the metric that they choose is the same as ours, i.e., the average clustering accuracy. In addition, the average clustering accuracy is commonly used in the works of NCD and clustering.
>
> >Q2. Further, it is not clear why k-means was chosen instead of other approaches such as cosine similarity, nearest neighbors etc.
>
> A2. Motivated by the empirical results shown in Figure 2b, we find that the existing NCD methods with deep clustering will seriously overfit to the noisy supervision (e.g., the pairwise similarity or the pseudo label) in the training procedures, bringing about the corresponding raise&fall accuracy curves. This is because of the powerful memory and learning capacity of deep neural networks. Thus, we act in a diametrically opposite way. Specifically, we detach the noisy supervision in time and propose a fully unsupervised clustering method, i.e., mini-batch prototypical K-means.
>
> Here we need to clarify a misuderstanding. Cosine similarity is only a similarity metric that is widely used in clustering methods, while not a clustering method. K-nearest neighbors is a supervised learning method requiring labeled data, which is not suitable for clustering problems.

---

### Decision · Program_Chairs · 2023-01-20

**Decision:**

Reject

**Justification For Why Not Higher Score:**

Given the issues mentioned above that still remain unaddressed, the paper does not warrant acceptance as is.

**Justification For Why Not Lower Score:**

N/A

**Metareview: Summary, Strengths And Weaknesses:**


This paper proposes the Novel Class Discovery under unreliable sampling (NUSA) problem, which is an extension of the vanilla NCD problem where labeled data may contain annotation errors and known classes may be confused with unknown classes. The paper develop several baselines based on prior works (tackling related but different problems) and then proposes a method that first trains on the data by considering all unlabeled instances as one class, and then a mini-batch prototypical k-means method. Results are shown across standard CIFAR-10/100 and ImageNet datasets.

  The reviewers appreciated the problem setting, which represents some practical challenges that may occur in the real world. They also noted that the method does seem to work compared to the baselines. However, they had several concerns that were raised, including: 1) Differences between some of the metrics reported for some of the baselines compared to published work (due to the difference between selection of best model versus model at final epoch) [g5Zb,SxkT], 2) Motivation and obfuscated description of the k-means based method [g5Zb,SxkT, 6g6E], and 3) lack of experiments on fine-grained datasets where the proposed problem of label noise would actually occur. While the authors provided a rebuttal, including some new experiments, the reviewers were not convinced and maintained their score. This is especially true of the last point, as that seems to best motivate the problem in the first place.

 Ultimately, while the setting is interesting and simple method is effective, there are too significant of issues related to the complex description of a simple method, and since it is largely a simple empirical approach the lack of comprehensive results on more challenging datasets (such as fine-grained ones) with clear analysis of which parts of the algorithm lead to successful performance. As a result, the paper does not reach the bar for acceptance, but the authors are recommended to address the above points for future resubmission.

**Summary Of Ac-Reviewer Meeting:**

N/A